# Comparison of the Effectiveness of Four Commercial DNA Extraction Kits on Fresh and Frozen Human Milk Samples

**DOI:** 10.3390/mps5040063

**Published:** 2022-07-19

**Authors:** Cassidy Butler, Amy Matsumoto, Casey Rutherford, Hope K. Lima

**Affiliations:** Department of Human Nutrition, Winthrop University, Rock Hill, SC 29733, USA; butlerc11@mailbox.winthrop.edu (C.B.); matsumotoa2@mailbox.winthrop.edu (A.M.); crutherford0918@gmail.com (C.R.)

**Keywords:** human milk, DNA extraction

## Abstract

For-profit donor human milk organizations have DNA-based proprietary methodology for testing incoming milk for adulteration with other species’ milk. However, there is currently no standardized methodology for extracting DNA from human milk. Microbiome research has shown that DNA purity and quantity can vary depending on the extraction methodology and storage conditions. This study assessed the purity and quantity of DNA extracted from four commercially available DNA extraction kits—including one kit that was developed for human milk. This study was for method validation only. One donor provided a 90 mL human milk sample. The sample was aliquoted into 70 × 1 mL microcentrifuge tubes. Aliquots were randomized into one of three categories: fresh extraction, extraction after freezing, and extraction after purification and storage at room temperature. DNA was analyzed for purity and quantity using a NanoDrop Spectrophotometer. Results confirmed differences in DNA purity and quantity between extraction kits. The Plasma/Serum Circulating DNA Purification Mini Kit (Norgen Biotek, ON, Canada) provided significantly more DNA, and consistent purity as measured by 260/280 and 260/230 ratios. DNA quantity and purity were similar between fresh and frozen human milk samples. These results suggest that DNA purity and quantity is highest and most consistent when extracted from human milk using the Plasma/Serum Circulating DNA Purification Mini Kit amongst the kits tested in this study. Standardized methodology for extracting DNA from human milk is necessary for improvement of research in the field of human milk. To do this, future studies are recommended for optimization of DNA extraction from human milk using larger sample sizes and multiple donor parents.

## 1. Introduction

The World Health Organization recommends exclusive human milk feeding for the first six months of life [1]; however, in the United States, only 45.7% of infants are meeting this recommendation [2] Barriers to exclusive human milk feeding include inadequate milk production, difficulty latching the infant, insufficient weight gain in the infant, confusion regarding milk–drug interactions, and inconsistent/lack of professional support [3,4,5] In many of these situations, it is in the best interest of the infant to continue to receive human milk to reduce or prevent health complications, especially when the infant is in the Neonatal Intensive Care Unit (NICU). When this occurs in the United States, accessing donor human milk through a milk bank or receiving shared human milk through informal channels is rising in popularity [6,7].

In the United States there are two models of milk banking: for-profit and not-for-profit. In not-for-profit models, individuals donating their milk are not compensated for providing milk. [8]. In many for-profit models of milk banking, donors are compensated based on the quantity of milk they provide [9,10]. Many breastfeeding activists feel that compensating individuals for their milk raises ethical and legal issues [11]. Of note is the concern that parents may mix their milk with milk of another species to increase the volume, thus increasing the compensation [12,13]. This can be detrimental to infants, due to the differing levels of protein and minerals in other animal-based milk [14,15]. Currently, one for-profit company reports that they screen donor milk for adulteration [16]; however, methods for screening are considered proprietary and no published protocols exist.

Product quality assurance efforts that have been established in milk banks are voluntary and have not been overseen by the FDA [17]. Not-for-profit milk banks in the United States operate under the Human Milk Banking Association of North America (HMBANA) guidelines. However, for-profit milk banks in the United States do not have standard operating procedures and protocols may differ between organizations. This raises concerns about product adulteration as screening is voluntary and protocols for detection are not approved or monitored by external agencies. Significant conflicts of interest arise when for-profit companies are self-monitoring, as profits depend on the sale and distribution of their products.

Efforts are ongoing to determine the best and most cost-effective way to test for human milk adulteration in milk banks. Some areas of research interest are in quantitative metabolomics, infrared spectroscopy, and DNA extraction with subsequent qPCR analysis [12,13,18]. Of particular interest in the development of methodologies for detecting human milk adulteration is DNA extraction with subsequent qPCR analysis. DNA extraction with subsequent qPCR analysis has a lower cost of equipment and sample analysis when compared with more sophisticated analytical techniques such a metabolomics or infrared spectroscopy. Many milk banks are limited in their ability to purchase research equipment, and as such, would not be able to integrate methodologies utilizing metabolomics or infrared spectroscopy into the day-to-day milk bank operations.

There are currently limitations to developing reliable and valid DNA analysis protocols for detection of adulteration in human milk. The first is that the most effective approach for isolating DNA from human milk at high concentrations and sufficient purities remains unknown. Current human milk microbiome research has shown that DNA purity and quantity can differ depending on extraction methodology, as well as between fresh and frozen milk samples [19,20]. Additionally, only one commercially available DNA extraction kit developed specifically for human milk is available [21]. Recent studies have utilized DNA extraction kits developed for plasma/serum, food, and soil to obtain DNA from human milk samples. In this study, we aim to compare the purity and quantity of DNA extracted from fresh or frozen human milk samples using four commercially available DNA extraction kits.

## 2. Materials and Methods

One 90 mL human milk sample was obtained from one mother. Seventy 600 μL aliquots were distributed into 1 mL microcentrifuge tubes. Of these 70 aliquots, 30 were frozen and 40 were left at room temperature (Figure 1). Thirty of the forty room temperature aliquots were immediately utilized for DNA extraction, performing 10 extractions per kit (Figure 1, Table 1). Ten of the forty room temperature aliquots were purified to allow for storage at room temperature using materials from the Milk DNA Preservation and Isolation kit. These 10 samples were then stored at room temperature for 3 weeks and subsequently, the remainder of the extraction process was completed. The DNA from frozen samples was extracted three weeks after freezing using three of the four DNA extraction kits, performing 10 extractions per kit (Figure 1, Table 1). The Milk DNA Preservation and Isolation Kit is only designed for use on human milk samples after preservation, and thus no frozen samples were utilized for DNA extraction with this kit.

All extractions were performed per the manufacturer’s instructions with the modification of beginning the extraction process with 200 μL of milk as an attempt to standardize the quantity of DNA extracted. A brief protocol description is provided in Table 1. Extracted DNA samples were analyzed in duplicate for purity (260/280 and 260/230 ratios) and quantity (ng/μL) using a NanoDrop 2000 (Thermo Scientific, Waltham, MA, USA). The 260/280 ratio is utilized in DNA analysis as a primary assessment of purity. The 260/280 ratio will be lower when the sample has high levels of protein or other contaminants that absorb at 280 nm. The 260/230 ratio is utilized in DNA analysis as a secondary measure of purity. The 260/230 ratio will be lower when the samples have high levels of carbohydrates or other contaminants that absorb at 230 nm.

### Statistical Analysis

Total nucleic acid (ng/μL), 260/280 ratio, and 260/230 ratio means (± SD) were calculated for each treatment group using Microsoft Excel 2016 (Redmond, WA, USA). Treatment means were compared in GraphPad (Prism, San Diego, CA, USA) using a Kruskal–Wallis one-way ANOVA with Dunn’s multiple comparison.

## 3. Results

### 3.1. 260/230. Ratio

Average 260/230 ratios are summarized in Figure 2. The Nucleospin Food Mini Kit for DNA from Food using fresh milk had the highest average 260/230 ratio; however, 260/230 ratios were not consistent between samples, as indicated by the large standard deviation. The E.Z.N.A.^®®^ Blood DNA Mini Kit and the Plasma/Serum Circulating DNA Purification Mini Kit had the most consistent 260/230 ratios and the 260/230 ratio did not vary significantly between the use of fresh or frozen milk.

### 3.2. 260/280. Ratio

Average 260/280 ratios are summarized in Figure 3. The Nucleospin Food Mini Kit for DNA from Food using frozen milk had the highest average 260/280 ratio; however, 260/230 ratios were not consistent between samples, as indicated by the large standard deviation. The Plasma/Serum Circulating DNA Purification Mini Kit had the second highest average 260/280 ratio, was consistent between samples, and did not vary significantly between the use of fresh or frozen milk.

### 3.3. Nucleic Acid Concentration

Average nucleic acid concentrations are summarized in Figure 4. The Plasma/Serum Circulating DNA Purification Mini Kit produced the highest nucleic acid concentration, and nucleic acid concentrations were consistent between samples. Slight differences were noted between fresh and frozen samples; however, the variation was not statistically significant.

## 4. Conclusions

Extracted DNA purity and quantity varied depending on which commercially available DNA extraction kit was utilized for sample analysis. Regarding DNA purity, the Milk DNA Preservation and Isolation Kit yielded the lowest purity as assessed by 260/230 and 260/280 ratios. The E.Z.N.A.^®®^ Blood DNA Mini Kit and the Plasma/Serum Circulating DNA Purification Mini Kit produced DNA with similar purity when compared with one another as assessed by the 260/230 and 260/280 ratios. Although the Nucleospin Food Mini Kit for DNA from Food had several samples with much higher purity than the other kits, the variance between samples was large, and thus would not provide extractions with consistent or reliable DNA purity. For the highest purity DNA, these data suggest utilizing the E.Z.N.A.^®®^ Blood DNA Mini Kit or Plasma/Serum Circulating DNA Purification Mini Kit.

The ability for the included DNA extraction kits to produce nucleic acid of appropriate purity is of particular interest. Both the 260/230 ratios and the 260/280 ratios were lower than is ideal for downstream qPCR analysis, which aims for a minimum of 2.0 and 1.8, respectively. The most consistent extraction kits did not even reach a 260/230 ratio of 1.0 in this experiment. The 260/280 ratios resulting from extraction with the E.Z.N.A ^®®^ Blood DNA Mini Kit and the Plasma/Serum Circulating DNA Purification Mini Kit were more favorable, but still less than ideal, with averages of 1.25 and 1.46, respectively. These low levels may be due to the high levels of fat present in human milk, which can negatively influence the effectiveness of DNA isolation buffers [22,23]. Both the Milk DNA Preservation and Isolation Kit and the Nucleospin Food Mini Kit for DNA from Food produced nucleic acid with qualities that would likely be inadequate for subsequent qPCR analysis.

Speaking to the DNA quantity, the Milk DNA Preservation and Isolation Kit and the Nucleospin Food Mini Kit for DNA from Food produced a significantly lower DNA quantity as assessed by nucleic acid concentration than the Plasma/Circulating DNA Purification Mini Kit. Although not statistically significant, DNA concentrations were higher when using the Plasma/Serum Circulating DNA Purification Mini Kit when compared with the E.Z.N.A.^®®^ Blood DNA Mini Kit. For the highest quantity of DNA, these results suggest utilizing the Plasma/Serum Circulating DNA Purification Mini Kit.

Nucleic acid concentration, 260/230 ratios, and 260/280 ratios were not statistically different when performing extraction on previously frozen human milk in any of the kits analyzed. This finding is different from data published in current human milk microbiome studies [20]. Future research is needed to confirm or deny the impact of freezing on DNA purity and quantity in human milk samples.

Taken together, these results suggest that DNA purity and quantity is highest when extracted using Plasma/Serum Circulating DNA Purification Mini Kit amongst the commercially available DNA extraction kits tested in this study. Additionally, these results suggest that utilizing the Plasma/Serum Circulating DNA Purification Mini Kit would allow for freezing of the human milk samples prior to analysis with no impact on DNA purity or quantity. It is a concern that the current DNA extraction kits that are intended specifically for human milk and food substances are the worst-performing kits, and better results are obtained using kits developed for plasma/serum. To continue to elevate the standard of human milk research that is performed, a reliable and valid means of extracting DNA from human milk needs to be developed and standardized throughout the literature.

Current literature comparing DNA extraction kits for use in human milk has focused on DNA extraction for use in microbiome analysis. Multiple studies have utilized the Milk DNA Preservation and Isolation Kit [19,23,24]; however, due to the focus on microbiome analysis, there is no mention of observed 260/280 ratios. One study by Cheema et al. reported DNA quantity from the Milk DNA Preservation and Isolation Kit of 0.68 ng/μL [24] from 1 mL of human milk, which is much lower than our reported output. Additionally, Cheema et al. reported that an average of only 44% of DNA extracted using the Milk DNA Preservation and Isolation kit was human DNA. This speaks to the importance of the DNA yield if the intended qPCR analysis is based on human DNA primers.

Another study by Lackey et al. reported DNA concentrations of 150 ng/μL [22], which was much higher than our reported output. Lackey et al. did include an optional two-hour enzymatic cell disruption with 10 μL lysozyme at 20 mg/mL. This step may be necessary for adequate cell disruption and DNA extraction. This study did not report the percentage of DNA that was human rather than microbial. In future studies, it may be necessary to include the optional cell disruption step and an analysis of how much of the resulting DNA was human DNA rather than microbial.

These data are an important first step in developing reliable and valid DNA extraction methodologies that can be utilized to create affordable protocols for detecting human milk adulteration in milk banks that provide compensation for human milk. These DNA extraction kits provide an efficient means of testing incoming human milk for adulteration with other species’ milk. The kits take about 30 min to run 24 samples and training for implementation is straightforward. Limitations of the pilot study that need to be addressed in future research include the sample size, the number of commercial DNA extraction kits included in the analysis, and the utilization of DNA extraction for subsequent qPCR analysis to confirm accurate detection of adulteration with other species’ milk. Given that our results show significant differences in purity and quantity of DNA extracted using commercially available DNA extraction kits, future studies are necessary to determine the ideal DNA extraction methodology for human milk research. Furthermore, once human milk DNA extraction methodologies are validated, the information needs to be tested to confirm sensitive and accurate detection of adulteration with other species’ milk is possible and realistic in a milk bank setting.

## Figures and Tables

**Figure 1 mps-05-00063-f001:**
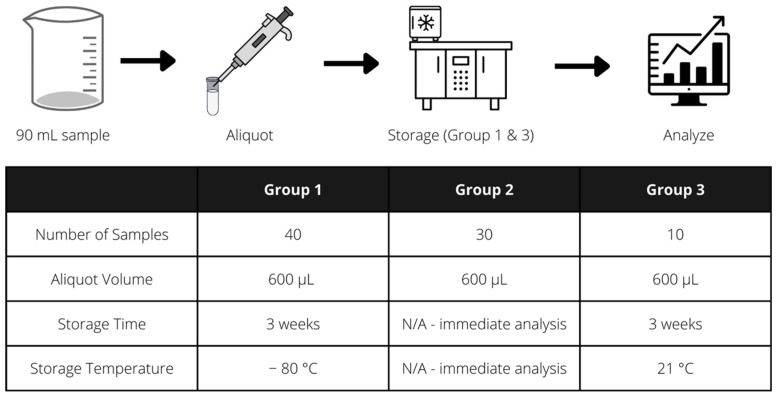
Summary of experimental process utilized to compare the purity and quantity of DNA extracted from four commercially available DNA extraction kits in fresh and frozen human milk samples.

**Figure 2 mps-05-00063-f002:**
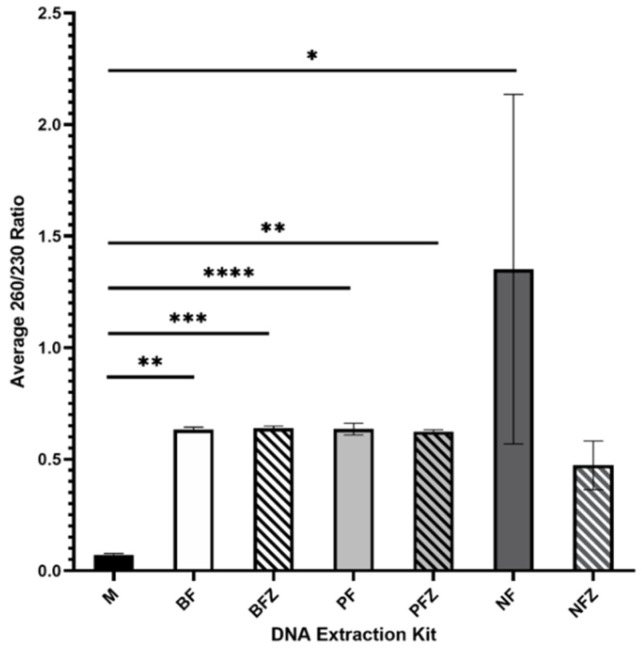
Average 260/230 ratio of DNA extracted from fresh (F) or frozen (FZ) human milk samples using Milk DNA Preservation and Isolation Kit (M), E.Z.N.A.^®®^ Blood DNA Mini Kit (B), Plasma/Serum Circulating DNA Purification Mini Kit (P), or the Nucleospin Food Mini Kit for DNA from Food (N); *n* = 10 for each treatment group. * indicates *p* < 0.05, ** indicates *p* < 0.01, *** indicates *p* < 0.001, **** indicates *p* < 0.0001.

**Figure 3 mps-05-00063-f003:**
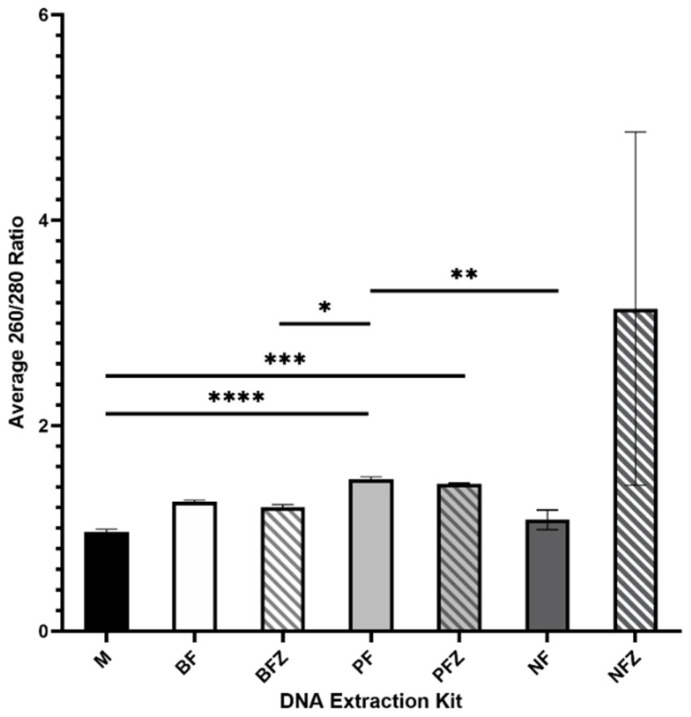
Average 260/280 ratio of DNA extracted from fresh (F) or frozen (FZ) human milk samples using the Milk DNA Preservation and Isolation Kit (M), E.Z.N.A.^®®^ Blood DNA Mini Kit (B), Plasma/Serum Circulating DNA Purification Mini Kit (P), or the Nucleospin Food Mini Kit for DNA from Food (N); *n* = 10 for each treatment group. * indicates *p* < 0.05, ** indicates *p* < 0.01, *** indicates *p* < 0.001, **** indicates *p* < 0.0001.

**Figure 4 mps-05-00063-f004:**
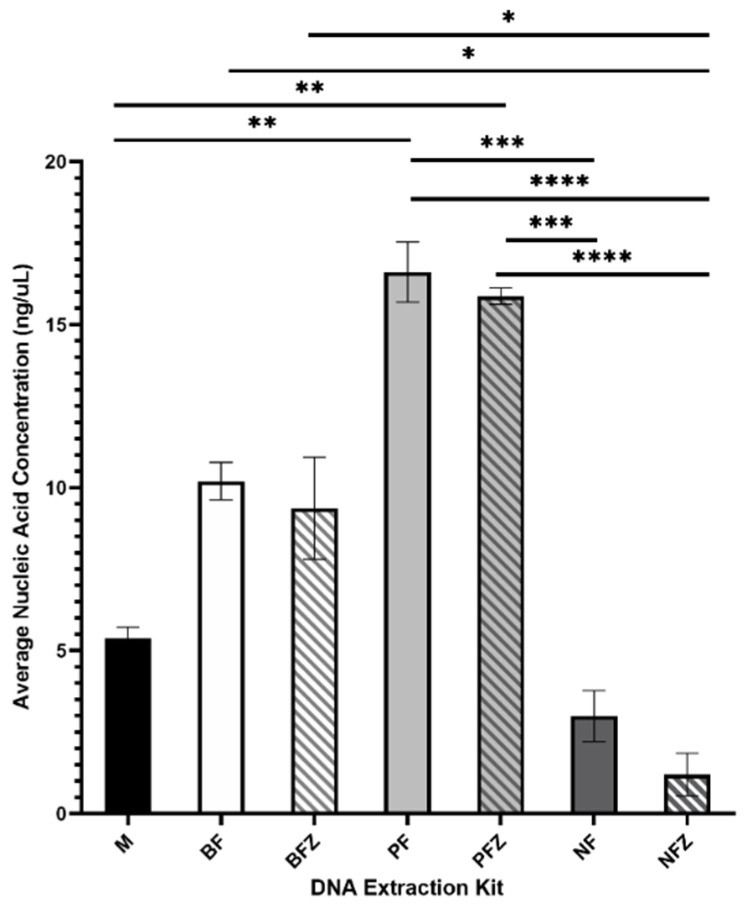
Average nucleic acid concentration of DNA extracted from fresh (F) or frozen (FZ) human milk samples using the Milk DNA Preservation and Isolation Kit (M), E.Z.N.A.^®®^ Blood DNA Mini Kit (B), Plasma/Serum Circulating DNA Purification Mini Kit (P), or the Nucleospin Food Mini Kit for DNA from Food (N); *n* = 10 for each treatment group. * indicates *p* < 0.05, ** indicates *p* < 0.01, *** indicates *p* < 0.001, **** indicates *p* < 0.0001.

**Table 1 mps-05-00063-t001:** Commercially available DNA extraction kits utilized for extraction of DNA from fresh and frozen human milk samples.

DNA Extraction Kit Name	Distributor	Catalog Number	Brief Protocol Description	Time to Completion	Samples Analyzed
Milk DNA Preservation and Isolation Kit (M)	Norgen Biotek Corp.	44,800	Initial preservation with proteinase K, purification additive, and isopropanol. Extraction with a silica-based spin column requiring binding, washing, and elution using a microcentrifuge and manufacturer-provided buffers.	80 min (20 min hands on)	10 fresh samples that had been purified and stored for 3 weeks at room temperature
E.Z.N.A.^®®^ Blood DNA Mini Kit (B)	Omega Biotek	D3392	Extraction with a silica-based column requiring binding, washing, and elution using a microcentrifuge and manufacturer-provided buffers.	30 min hands on	10 fresh and 10 frozen
Plasma/Serum Circulating DNA Purification Mini Kit (P)	Norgen Biotek Corp.	50,600	Extraction using a resin-based slurry to attract DNA followed by washing elution utilizing manufacturer provided elution buffer.	30 min hands on	10 fresh and 10 frozen
NucleoSpin Food Mini Kit for DNA from Food (N)	Macherey-Nagel	740,945.50	Extraction with a silica-based spin column requiring binding, washing, and elution using a microcentrifuge and manufacturer-provided buffers.	30 min/6 preps hands on	10 fresh and 10 frozen

## Data Availability

The data collected in this study is all contained in the article.

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
