# Peer review of "Comparison of the Effectiveness of Four Commercial DNA Extraction Kits on Fresh and Frozen Human Milk Samples"

_mps, 2022, doi:10.3390/mps5040063_

Round 1
Reviewer 1 Report
In this manuscript, the authors compared the quantity and quality of DNA obtained from four commercial DNA extraction kits to identify the best DNA extraction kit for the processing of fresh or frozen human milk samples for quality control testing for human milk adulteration. Analyses of the 260/230, 260/280 ratios and the averaged nucleic acid concentration obtained from 200uL volumes of human milk extracted by the four kits showed that the Plasma/Serum Circulating DNA Purification Mini kit returned DNA samples of the highest DNA and quality.
The authors have submitted this manuscript to be reviewed as an original research article which requires “scientifically sound experiments and provides a substantial amount of new information” as stated in the MPs Instructions for authors, however, the overall information presented is too brief and fails to deliver on both counts. As expressed in the introduction by the authors, there is currently no standardized protocol for the screening of milk adulteration. Therefore, the evaluation of different DNA extraction kits for the human milk can be useful and important for human milk banking laboratories. However, there is also no discussion of the results apart from brief statements in the conclusion, which are anecdotal at best, and no references to put the presented results in context with the available literature, or any demonstration of the significance of the results generated is new information.
While the authors did state that the objectives of this study were for method validation only, and aim to only compare the quality and quantity of DNA extracted from fresh or frozen milk samples, the experimental design should include tests of sensitivity, specificity, reproducibility (partly addressed with 10 replicates) and repeatability.
Some things to consider -
Are there any quality assurance standards relevant to human milk storage that milk banks have to meet currently?
Comment and discuss the differences between the four kits? Solid phase / magnetic bead based / organic extraction? Cost of the kit? What is the total time taken for each protocol? Is it acceptable for the workflow in a milk bank?
The kit may return the highest quality nucleic acid quantity and quality, but is it able to extract mixtures of milk from human and other animals effectively to ensure that downstream adulteration tests are accurate? Given that lipid content can have an effect on DNA extraction efficiency, and differences in fat content in milk between humans and other animals.
Other comments:
Maintain consistency with the units of measurement for milk, whether in imperial or metric units of volume.
In Figure 1, the microlitre was presented as uL instead of µL.
Author Response
Reviewer 1:
The authors would like to thank the reviewer for their time and expertise in providing feedback to us for improvement of our manuscript.
However, there is also no discussion of the results apart from brief statements in the conclusion, which are anecdotal at best, and no references to put the presented results in context with the available literature, or any demonstration of the significance of the results generated is new information. The conclusions section has been expanded to put the results in context with the available literature and the possible applications it may have in human milk screening in donor milk banks.
While the authors did state that the objectives of this study were for method validation only, and aim to only compare the quality and quantity of DNA extracted from fresh or frozen milk samples, the experimental design should include tests of sensitivity, specificity, reproducibility (partly addressed with 10 replicates) and repeatability. It is the intention of the authors to expand upon these preliminary results with future research. A limitations section has been added to address these concerns.
Some things to consider -
- Are there any quality assurance standards relevant to human milk storage that milk banks have to meet currently? For-profit donor milk banks do not have quality assurance standards that they have to meet and have not been overseen by the FDA. I have updated the introduction to explicitly state this and the concerns it raises.
- Comment and discuss the differences between the four kits? Solid phase / magnetic bead based / organic extraction? Cost of the kit? What is the total time taken for each protocol? Is it acceptable for the workflow in a milk bank? Table 1 has been updated to include additional kit details, including a brief protocol and time to completion. The conclusion has been updated to contain discussion about application in a milk bank setting.
- The kit may return the highest quality nucleic acid quantity and quality, but is it able to extract mixtures of milk from human and other animals effectively to ensure that downstream adulteration tests are accurate? Given that lipid content can have an effect on DNA extraction efficiency, and differences in fat content in milk between humans and other animals. This is the next step in our research and this has been added in the limitations/future studies paragraph in the conclusions section of the paper.
- Other comments:
- Maintain consistency with the units of measurement for milk, whether in imperial or metric units of volume. Units have been updated throughout the manuscript
- In Figure 1, the microlitre was presented as uL instead of µL. This error has been corrected
Reviewer 2 Report
This work is about the comparison of four commercial DNA extraction kit for human milk samples. Comparison was simply described by three terms – A260/A280, A260/A230, and DNA concentration. Simple and clear results are quite good to obtain useful information, and could be suitable in short reports in Methods and Protocols. After major revision about below comments, I believe it would be published.
1. Authors use ‘Ounce’ unit for total volume of samples. In the whole paper (especially short report), volume unit should be unified. Please change it to SI unit (mL or μL).
2. Figure 1 is summary of experimental process. To show more information clearly, please add simple scheme or schematic description, instead of text only. A few of simple clipart (easily usable after Google search, such as beaker, small bottle, pipette, etc.) may be useful.
3. Are all used DNA extraction kits, M, B, P, N, silica-column system? Please provide common protocol. If there are more information about difference of each solution type, that will be great for readers.
4. Graphs contained large scale STDEV, such as NF in Figure 2 and NFZ in Figure 3. Is it reliable? Simple investigation about kit N (e.g. salt composition of ‘kit N buffer’ may bring this scale of variation) could help this discussion.
5. In terms of DNA extraction, storage condition of human milk (fresh or frozen) doesn’t matter, right?
6. There are some typos. Please revise the manuscript before the final submission. I found these, but there are more.
Page 3, line 84 // ( SD)
Page 4, line 97, Figure 2 legend // Table 10
Page 4, line 97, Figure 2 legend // Abbreviation info has been skipped only this figure.
Author Response
Reviewer 2:
The authors would like to thank the reviewer for their time and expertise in providing feedback to us for improvement of our manuscript.
- Authors use ‘Ounce’ unit for total volume of samples. In the whole paper (especially short report), volume unit should be unified. Please change it to SI unit (mL or μL). Volumes have been updated throughout the manuscript to mL
- Figure 1 is summary of experimental process. To show more information clearly, please add simple scheme or schematic description, instead of text only. A few of simple clipart (easily usable after Google search, such as beaker, small bottle, pipette, etc.) may be useful. The image has been updated to a graphic/table combination
- Are all used DNA extraction kits, M, B, P, N, silica-column system? Please provide common protocol. If there are more information about difference of each solution type, that will be great for readers. – Table 1 has been updated to have a brief description; many of the solutions are general “additives” provided in the kits so no additional solution information was added to address solutions.
- Graphs contained large scale STDEV, such as NF in Figure 2 and NFZ in Figure 3. Is it reliable? Simple investigation about kit N (e.g. salt composition of ‘kit N buffer’ may bring this scale of variation) could help this discussion. Conclusions have been updated to include a discussion about reliability of kit N, which had a large STDEV
- In terms of DNA extraction, storage condition of human milk (fresh or frozen) doesn’t matter, right? The development of this study was based on observation within our lab of differing DNA quality/quantity utilizing the food kit after freeze/thaw cycles – we wanted to expand on this with a comparison trial to verify that the differences were kit-based rather than an issue of storage with the human milk. One study was located in microbiome research that showed a difference, it has been added as a reference.
- There are some typos. Please revise the manuscript before the final submission. I found these, but there are more.
Page 3, line 84 // ( SD)
Page 4, line 97, Figure 2 legend // Table 10
Page 4, line 97, Figure 2 legend // Abbreviation info has been skipped only this figure.
Manuscript has been reviewed for typos and they have been corrected.
Reviewer 3 Report
This study presents a comparison of DNA extraction performances from four commercial kits. The authors extracted DNA from a human milk sample from one donor and compared the extraction performances in qualitative and quantitative ways. Those findings in this study are expected to provide a standardized methodology for DNA extraction in the field of human milk research. The manuscript is well-structured, with excellent reasonable protocols, data collection, and analysis. There are some comments and/or questions as follows.
(1) Lines 46-47: what is the source of this report?
(2) Introduction section: regarding the human milk adulteration, reference number 9 was introduced. More previous studies in this field and state-of-the-art are needed to highlight the importance of this study.
(3) The protocol for kit M: the authors did not extract DNA immediately for kit M while they did for other kits. Is there any reason for this?
(4) Qualitative measures: information on 260/280 and 260/230 ratios would be helpful. Especially, since the authors take these measures with linearity, references for this linearity seem to be necessary.
(5) The results from the four kits are interesting. However, the contributions of this study in this field of research need to be addressed explicitly.
(6) Some presentation of the limitation of this study would be helpful, including as single source of samples
Sincerely,
The reviewer.
Author Response
Reviewer 3:
The authors would like to thank the reviewer for their time and expertise in providing feedback to us for improvement of our manuscript.
(1) Lines 46-47: what is the source of this report? The authors apologize, but we do not have line numbers on the version of the manuscript that we were provided. I think that it may be referring to the report that one for-profit company screens their milk. We have added a citation to this assuming it was correct. If this is not the correct item, can you please provide more guidance on which statement?
(2) Introduction section: regarding the human milk adulteration, reference number 9 was introduced. More previous studies in this field and state-of-the-art are needed to highlight the importance of this study. – the introduction has been modified to include an overview of the other state-of-the-art technologies that are being utilized to investigate human milk adulteration.
(3) The protocol for kit M: the authors did not extract DNA immediately for kit M while they did for other kits. Is there any reason for this? – Yes, this kit is designed to have a preservation step that is utilized immediately to make the samples stable at room temperature for future analysis. We utilized the kit with the manufacturer instructions so therefore, did not skip the preservation step. The manuscript has been updated in the methodology to make this clear.
(4) Qualitative measures: information on 260/280 and 260/230 ratios would be helpful. Especially, since the authors take these measures with linearity, references for this linearity seem to be necessary. The 260/280 and 260/230 ratios are more accurately a measure of purity rather than quality, and the manuscript has been updated to reflect this. Information has also been added to the methods to describe the meaning of the ratios.
(5) The results from the four kits are interesting. However, the contributions of this study in this field of research need to be addressed explicitly. The introduction and conclusions have been expanded to explicitly state the application in the field.
(6) Some presentation of the limitation of this study would be helpful, including as single source of samples The conclusions section has been expanded to include an assessment of limitations and proposal of future research.
Round 2
Reviewer 1 Report
This manuscript looks at the performance of four DNA extraction kits on fresh and frozen milk human samples. The additional information in the introduction has certainly helped to highlight the importance of standardising procedures and improving quality assurance protocols for human milk banking. The addition of Figure 1 and the information of the different DNA extraction kits were an improvement and helped to understand what was being compared.
As mentioned in the first review, this manuscript is intended as an original article, and the authors need to bring forward the new knowledge and novel results that is demonstrated in this study. The Milk DNA Preservation kit, and the Nucleospin food DNA kits which has been designed for the specific biological samples are the worst performing kit, while the EZDNA and plasma/serum kits, which are intended for other liquid biological samples such as blood and serum are the best performing. This in my opinion is the significant take home “new” finding and should be discussed in more detail.
The study is evaluating commercial DNA extraction kits for processing human milk, but the authors have not discussed the results obtained in context with published studies that have evaluated similar DNA extraction kits or other DNA extraction methods for milk. There is only one reference added to the discussion, and if the authors are referring to current human milk microbiome studies, at least two references should be provided. Furthermore, for a manuscript comparing different DNA extraction kits for milk, the authors have not referenced any related DNA extraction studies.
· Correct and update the reference numbering in the Introduction, 4th paragraph “…..subsequent qPCR analysis. 18, 189 12”
· In Materials and Methods, 30mLs of human milk is not enough for 70 aliquots of 600microlitre volumes. In Figure 1, the diagram shows 3 ounce sample which is ~90mLs, can the authors update the units of volume, correct amount of sample collected and the number of aliquots prepared.
· A 260/230 ratio of 2.0-2.2 is ideal for nucleic acid purity, the authors should comment and discuss why the results from the kits are so much lower than the 2.0 ratio. It is especially low for the milk DNA preservation kit, and this should be mentioned in the results and discussed (see comments below).
· A 260/280 ratio of 1.8 is ideal for pure DNA, and this was most closely met by EZDNA and Plasma/serum kit, the milk DNA preservation and nucleospin did not perform as well and this should be indicated in the results and discussed.
A quick search has brought up two studies which have also used the Milk DNA preservation kit and also other studies which looked at DNA extraction from milk which may be useful for comparing the results in this study. There are many more but here are some suggestions. None of the kits are specific for human genomic DNA, so a lot of useful information can be gleaned and used to discuss the results here even if they were dairy and bovine milk studies:
Lackey KA, Williams JE, Price WJ, Carrothers JM, Brooker SL, Shafii B, McGuire MA, McGuire MK. Comparison of commercially-available preservatives for maintaining the integrity of bacterial DNA in human milk. J Microbiol Methods. 2017 Oct;141:73-81. doi: 10.1016/j.mimet.2017.08.002. Epub 2017 Aug 9. PMID: 28802721.
yons KE, Fouhy F, O' Shea CA, Ryan CA, Dempsey EM, Ross RP, Stanton C. Effect of storage, temperature, and extraction kit on the phylogenetic composition detected in the human milk microbiota. Microbiologyopen. 2021 Jan;10(1):e1127. doi: 10.1002/mbo3.1127. Epub 2020 Dec 29. PMID: 33373099; PMCID: PMC7841076.
- (2020) Extraction and detection of DNA from UHT milk during storage, CyTA - Journal of Food, 18:1, 747-752, DOI: 10.1080/19476337.2020.1839565
- Murphy, Michael & Shariflou, Mohammad & Moran, Chris. (2002). High quality genomic DNA extraction from large milk samples. The Journal of dairy research. 69. 645-9. 10.1017/S0022029902005848.
- Volk, Helena & Piskernik, Saša & Kurinčič, Marija & Klancnik, Anja & Toplak, Nataša & Jeršek, Barbara. (2013). Evaluation of different methods for DNA extraction from milk. Journal of food and nutrition research. ISSN 1336-8672. 1-10 (ACCEPTED).
- T. Usman*, Y. Yu*, C. Liu, Z. Fan and Y. Wang Genet. Mol. Res. 13 (2): 3319-3328 (2014) Comparison of methods for high quantity and quality genomic DNA extraction from raw cow milk
- Psifidi A, Dovas CI, Banos G. A comparison of six methods for genomic DNA extraction suitable for PCR-based genotyping applications using ovine milk samples. Mol Cell Probes. 2010 Apr;24(2):93-8. doi: 10.1016/j.mcp.2009.11.001. Epub 2009 Nov 10. PMID: 19900535.
- Pokorska J, Kułaj D, Dusza M, Żychlińska-Buczek J, Makulska J. New Rapid Method of DNA Isolation from Milk Somatic Cells. Anim Biotechnol. 2016;27(2):113-7. doi: 10.1080/10495398.2015.1116446. PMID: 26913552; PMCID: PMC4806335.
Author Response
Thank you for your time in completing the review of our manuscript. Please see attached file for responses.

Reviewer 2 Report
Authors revised the manuscript very well. I feel current form is suitable to methods and protocols journal, and helpful for readers of this journal. It's my pleasure to contribute a little bit.
Author Response
Thank you for your time in reviewing our manuscript. Please see attached file for our response.

Round 3
Reviewer 1 Report
The added discussion points in context with the results and related references have significantly improved the manuscript.
There are minor typos which need to be corrected:
In the abstract, the volume of milk collected from the donor needs to be updated from 30mL to 90mLs.
All registered trademark symbols should be superscripted.
In Figure 1, the volume needs to be updated from 3 ounce sample to 90mLs sample.
Author Response
Please see attached

This manuscript is a resubmission of an earlier submission. The following is a list of the peer review reports and author responses from that submission.